# Systematic investigation of the link between enzyme catalysis and cold adaptation

Catherine Stark[1,2], Teanna Bautista-Leung[2], Joanna Siegfried[2], Daniel Herschlag[1,2,3]*

[1]ChEM-H, Stanford University, Stanford, United States; [2]Department of Biochemistry, Stanford University, Stanford, United States; [3]Department of Chemical Engineering, Stanford University, Stanford, United States

**Abstract** Cold temperature is prevalent across the biosphere and slows the rates of chemical reactions. Increased catalysis has been predicted to be a dominant adaptive trait of enzymes to reduced temperature, and this expectation has informed physical models for enzyme catalysis and influenced bioprospecting strategies. To systematically test rate enhancement as an adaptive trait to cold, we paired kinetic constants of 2223 enzyme reactions with their organism's optimal growth temperature ($T_{Growth}$) and analyzed trends of rate constants as a function of $T_{Growth}$. These data do not support a general increase in rate enhancement in cold adaptation. In the model enzyme ketosteroid isomerase (KSI), there is prior evidence for temperature adaptation from a change in an active site residue that results in a tradeoff between activity and stability. Nevertheless, we found that little of the rate constant variation for 20 KSI variants was accounted for by $T_{Growth}$. In contrast, and consistent with prior expectations, we observed a correlation between stability and $T_{Growth}$ across 433 proteins. These results suggest that temperature exerts a weaker selection pressure on enzyme rate constants than stability and that evolutionary forces other than temperature are responsible for the majority of enzymatic rate constant variation.

*For correspondence: herschla@stanford.edu

Competing interest: The authors declare that no competing interests exist.

## Editor's evaluation

Are enzymes found in organisms that optimally grow at colder temperature are more active than the same enzymes found in organisms that optimally grow at warmer temperatures? Here, an assessment of the catalytic constants for approximately 2200 enzymes (obtained from the BRENDA database) showed no correlation between the relative catalytic activity and the optimum growth temperature. Further support for this conclusion was obtained from the measurement of the catalytic constants from a selection of ketosteroid isomerases from organisms that optimally grow between 15 and 46 degrees centigrade. Therefore, additional factors must dominate the selection pressure used to evolve enzymes of the appropriate catalytic activity. In contrast, growth temperature exerts a significant selective pressure on protein stability.

## Introduction

Temperature is a ubiquitous environmental property and physical factor that affects the evolution of organisms and the properties and function of the molecules within them. As reaction rates are reduced at lower temperatures (*Arrhenius, 1889*; *Wolfenden et al., 1999*), the maintenance of enzyme rates has been suggested to be a universal challenge for organisms at colder temperatures that do not regulate their internal temperature (*D'Amico et al., 2003*; *Fields et al., 2015*; *Siddiqui*

*and Cavicchioli, 2006*; *Zecchinon et al., 2001*). According to what we term the rate compensation model of temperature adaptation, this challenge has been suggested to be met by cold-adapted enzyme variants providing more rate enhancement than the corresponding warm-adapted variants (*Figure 1A*). This model predicts that cold-adapted variants are generally faster than warm-adapted variants when assayed at a common temperature (*Figure 1B*). Indeed, this behavior has been reported for diverse enzymes, and these observations have been taken as support for this model (*Figure 1C and D*; *Collins and Gerday, 2017*; *Feller and Gerday, 1997*; *Siddiqui and Cavicchioli, 2006*; *Smalås et al., 2000*).

The observed effects on enzymatic rate constants (*Figure 1C and D*) have also led to proposals of general physical models for cold adaptation linked to flexibility, as outlined in *Feller and Gerday, 2003*; *Fields et al., 2015*; *Åqvist et al., 2017*; *Arcus et al., 2016*; *Nguyen et al., 2017*; *Saavedra et al., 2018*. Further, features identified in comparative structural analyses of cold- and warm-adapted enzymes, such as fewer surface hydrogen bonds and salt bridges (*Cai et al., 2018*), have been suggested to increase flexibility and thereby increase catalysis (*Mandelman et al., 2019*; *Park et al., 2018a*; *Park et al., 2018b*). Correspondingly, the study of cold adaptation may have the potential to provide generalizable insights into physical properties of enzymes that make them better catalysts, a longstanding challenge in the field (*Blow, 2000*; *Hammes et al., 2011*; *Kraut et al., 2003*; *Ringe and Petsko, 2008*). From a practical perspective, the prediction of enhanced catalysis by cold-adapted enzymes has motivated low-temperature bioprospecting for biocatalysts to use in industrial processes (*Bhatia et al., 2021*; *Bruno et al., 2019*; *Kuddus, 2018*; *Santiago et al., 2016*).

Given the theoretical and practical implications of the proposed relationship between enzyme rate enhancement and organism growth temperature, we sought to test the generality of the rate compensation model of temperature adaptation. We collated enzyme rate constant data (*Chang et al., 2021*) and organism optimal growth temperature ($T_{Growth}$) (*Engqvist, 2018*) for 2223 reactions using public databases. The results revealed no enrichment of faster reactions with colder growth temperatures, and thus did not support increased rate enhancement with decreasing environmental temperature as a prevalent adaptation in nature. Further, we found that most rate constant variation for the enzyme ketosteroid isomerase (KSI) is not accounted for by $T_{Growth}$ despite strong evidence for temperature adaptation within its active site (*Pinney et al., 2021*). In contrast, a similar broad analysis revealed that stability correlates with $T_{Growth}$, as expected. Our results suggest that temperature exerts a weaker selection pressure on enzyme rate enhancement than stability and that other evolutionary forces are responsible for most variation in enzymatic rate enhancements.

## Results
### Systematically testing the rate compensation model

To investigate temperature adaptation of enzyme rate enhancement, we paired rate constant data from the BRENDA database (*Chang et al., 2021*) to organism growth temperatures. We simplified organism temperatures that may span changing conditions (*Doblin and van Sebille, 2016*) by matching the species name associated with the enzyme variant with the organism optimal growth temperature ($T_{Growth}$) (*Engqvist, 2018*). Of 76,083 $k_{cat}$ values in BRENDA, we found that 49,314 were for wild-type enzymes. Of these data, 16,543 values matched to microorganisms with known $T_{Growth}$ values. We selected reactions in the database with variants from more than one organism, spanning 7086 $k_{cat}$ values for 2223 reactions across 815 organisms with at least two variants per reaction (*Figure 2A*). These reactions spanned a temperature range of 1–83°C (*Figure 2B*).

For each enzyme reaction, we first calculated the rate ratio ($k_{cold}/k_{warm}$) between the rate constant of the variant from the lowest growth temperature organism and the rate constant of the variant from the highest growth temperature organism. We observed rate ratios greater than one (1142 reactions) as predicted by rate compensation, but nearly the same number of rate ratios of less than one (1082 reactions) (*Figure 2C*, compare with *Figure 1D*), providing no support for widespread or predominant rate compensation.

To assess whether rate constant trends were obscured by mixed assay temperatures or narrow $T_{Growth}$ ranges, we analyzed the distributions of rate ratios separated by assay temperature (25°C or 37°C; *Figure 2—figure supplement 1A and B*) and the rate ratios for data representing wider $T_{Growth}$

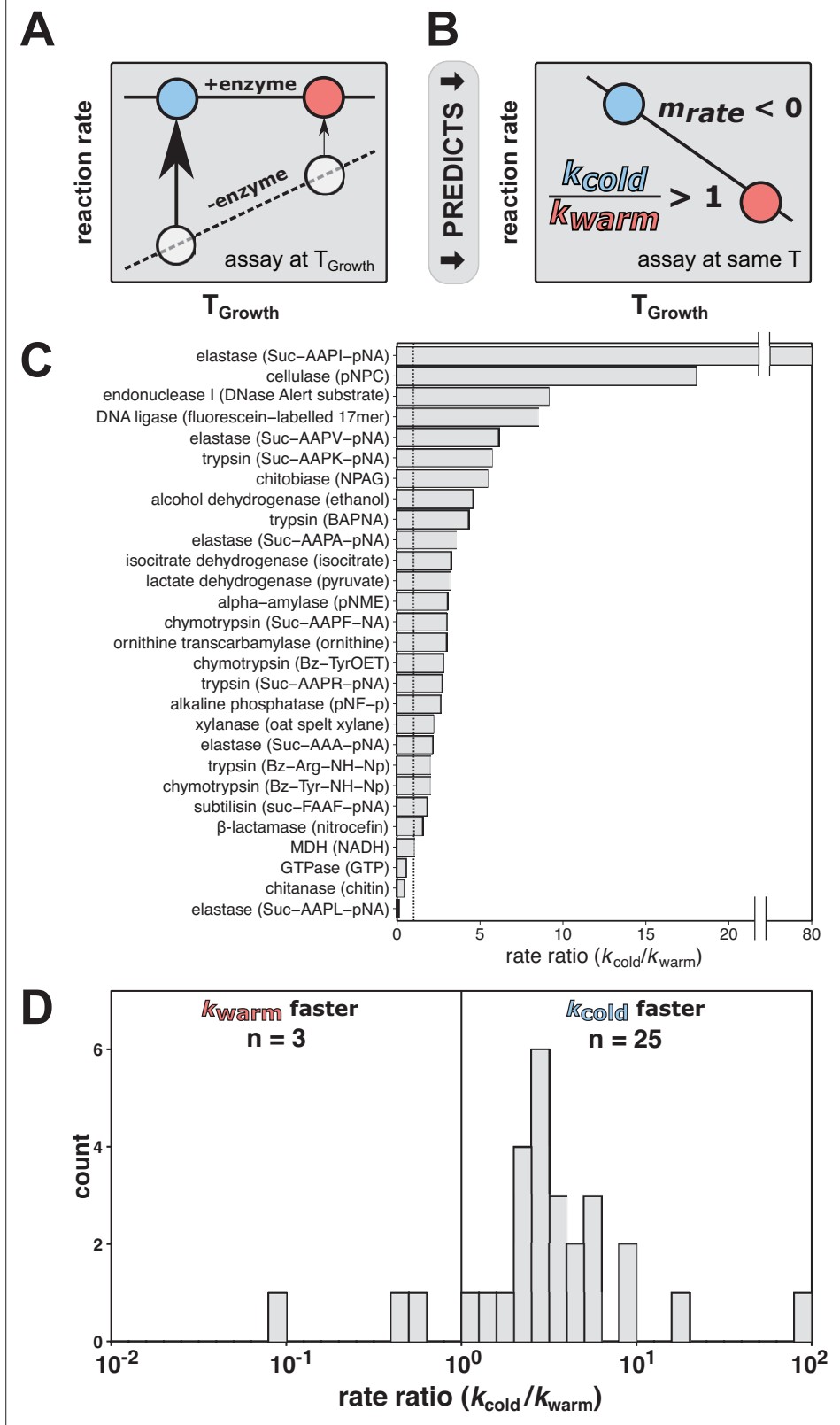

**Figure 1.** The rate compensation model of cold adaptation predicts that cold-adapted enzymes exhibit greater catalysis and are faster at a common temperature than their warm-adapted counterparts. (**A**) According to the rate compensation model of cold adaptation, a cold-adapted variant (blue circle) has larger rate enhancement than a warm-adapted variant (red circle). The dashed line represents the uncatalyzed reaction, the solid line represents

*Figure 1 continued on next page*

*Figure 1 continued*

the catalyzed reaction, and the arrows represent the rate enhancement at the respective organism $T_{Growth}$. **(B)** When variants are assayed at a common temperature, rate compensation predicts a faster reaction for the enzyme from the cold-adapted organism, corresponding to a rate ratio ($k_{cold}/k_{warm}$) of greater than one and a negative slope of rate vs. $T_{Growth}$ ($m_{rate}$). **(C, D)** Rate comparisons of warm-adapted and cold-adapted enzyme variants made at identical temperatures from cold adaptation literature spanning indicated reactions with substrate specified in parentheses (*Collins and Gerday, 2017*; *Feller and Gerday, 1997*; *Siddiqui and Cavicchioli, 2006*; *Smalås et al., 2000*). The black vertical lines represent no rate enhancement change with temperature (i.e., rate ratio = 1).

The online version of this article includes the following source data for figure 1:

**Source data 1.** Rate comparisons of warm-adapted and cold-adapted enzyme variants made at identical temperatures from cold adaptation literature.

---

ranges (>Δ20°C or >Δ60°C; *Figure 2—figure supplement 1C and D*). No temperature-dependent trends emerged, supporting the above conclusion of an absence of widespread rate compensation.

To derive a control distribution, we compared enzyme variant rate constants originating from different organisms with identical $T_{Growth}$ values. We found 615 reactions with more than one variant assigned the same $T_{Growth}$, and we calculated the rate ratio and its reciprocal ($k_{max}/k_{min}$ and $k_{min}/k_{max}$) for each reaction. This control distribution (dashed line, *Figure 2D*) was indistinguishable from the data distribution of rate ratios across $T_{Growth}$ (solid line, *Figure 2D*; p = 0.21, Mann–Whitney U test, two-sided). Analogous analyses of $k_{cat}/K_M$ values lead to the same conclusions (*Figure 2—figure supplement 2*).

As it is not possible to prove the absence of a relationship (*Altman and Bland, 1995*), we examined the slope ($m_{rate}$) of $k_{cat}$ values vs. $T_{Growth}$ for each of the 951 reactions with >2 variants (*Figure 2D and E*, *Figure 2—figure supplement 3*) to address whether there might be a limited set of enzyme reactions exhibiting significant cold adaptation through a mechanism of enhanced rate. We found two reactions (triose-phosphate isomerase with glyceraldehyde 3-phosphate and cutinase with 4-nitrophenyl butyrate) significantly but *positively* associated with $T_{Growth}$ (Bonferroni correction; p-value < $5.3 \times 10^{-5}$, n = 951).

In summary, the data provide no indication of increased rate enhancements as a consequence of decreasing $T_{Growth}$. These results suggest that rate compensation is not a universal or prevalent consequence of temperature adaptation. The prior conclusion of widespread temperature adaptation may have arisen from the use of a small set of enzymes (n = 28; *Figure 1C and D*) or from inadvertent confirmation bias (*Nickerson, 1998*).

## Testing the rate compensation model for the enzyme KSI

To probe rate compensation in greater depth, we turned to the enzyme KSI for which recent data has demonstrated rate compensation (*Pinney et al., 2021*). Specifically, the change of a single active site residue at position 103 from serine (S103, prevalently found in warm-adapted KSI variants) to protonated aspartic acid (D103, prevalently found in mesophilic KSI variants) provided improved activity from a stronger hydrogen bond while also sacrificing stability by introducing an unfavorable protonation coupled to folding. We therefore used KSI to more deeply investigate the potential for rate compensation by assaying 20 variants that vary in sequence and $T_{Growth}$ (*Figure 3A*).

KSI catalyzes the double bond isomerization of steroid substrates (*Figure 3B*) and is predicted to be part of a pathway that enables degradation of steroids for energy and carbon metabolism in bacteria (*Horinouchi et al., 2010*). KSI variants were identified by sequence relatedness to known KSIs. The 20 selected KSI variants ranged between 20% and 75% sequence identity to each other (*Figure 3—figure supplement 1*) and were selected from bacteria originating from environments spanning glaciers, ocean floor, soil, and wastewater with reported $T_{Growth}$ values from 15°C to 46°C (*Figure 3—source data 1*). Each purified KSI demonstrated similar circular dichroism (CD) spectra at 5°C and 25°C, suggesting that variants were not unfolding at the 25°C assay temperature (*Figure 3—figure supplement 2*). All putative KSI variants exhibited isomerase activity on the steroid substrate 5 (10)-estrene-3,17-dione (5 (10)-EST) (*Figure 3C*).

We observed that the KSIs with the prevalent cold-adapted residue (D103 and the similar residue E103, *Pseudomonas putida* numbering) were not uniformly faster than other KSIs in $k_{cat}$ (*Figure 3C*) or $k_{cat}/K_M$ (*Figure 3—figure supplement 3*). The observation that one of the fastest variants contained

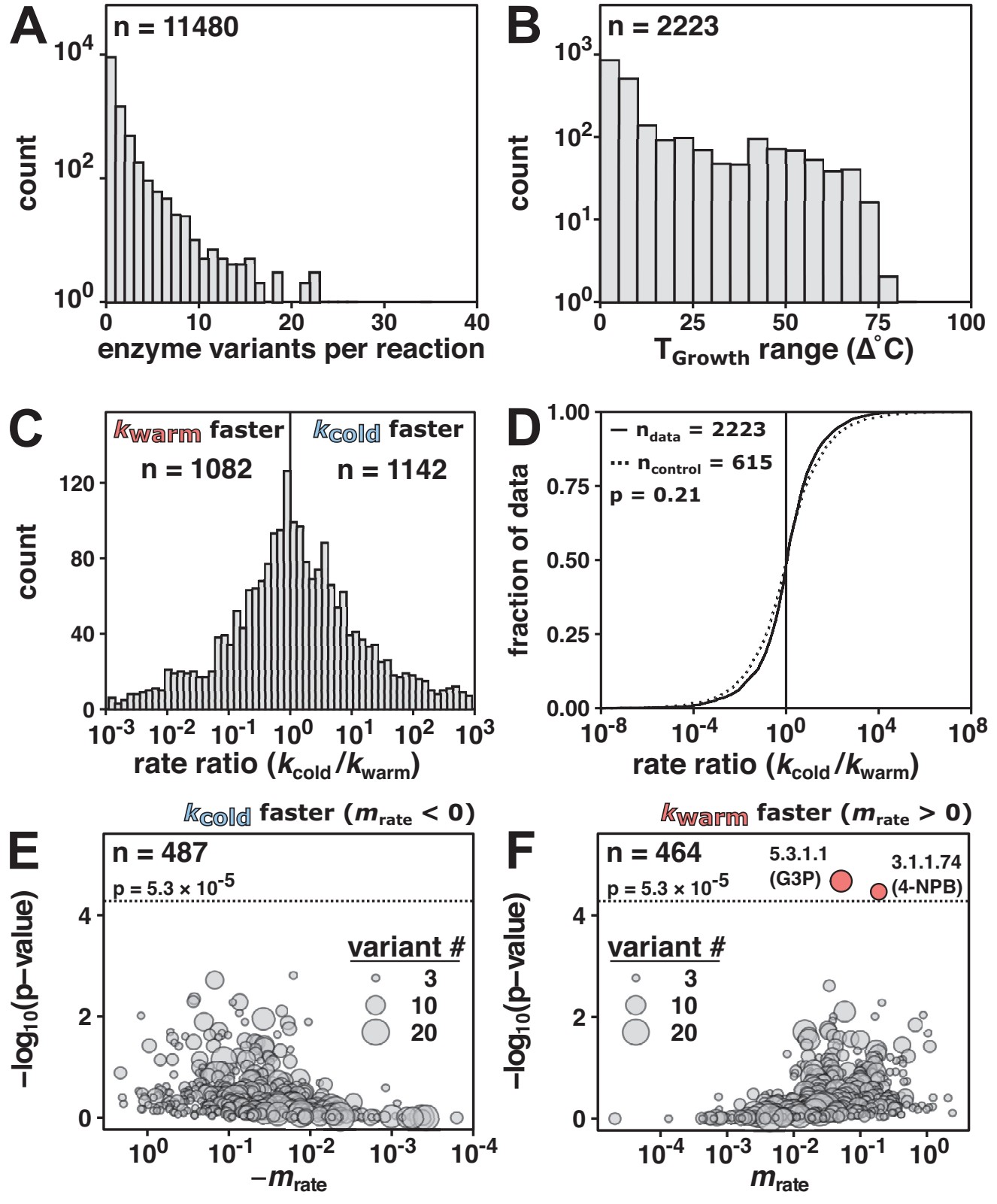

**Figure 2.** Enzyme rate constant data ($k_{cat}$) do not indicate general rate compensation. (**A**) Enzyme variants per reaction of wild-type enzyme $k_{cat}$ values ($n$ = 11,480 reactions) matched to $T_{Growth}$. (**B**) Reactions with more than one enzyme variant ($n$ = 2223 reactions). (**C**) Rate ratio distribution of the rate at the coldest $T_{Growth}$ ($k_{cold}$) divided by the rate of the variant from the warmest $T_{Growth}$ ($k_{warm}$) (median = 1.1-fold, 95% CI [1.00, 1.22], $n$ = 2223 reactions). Vertical line at rate ratio = 1. For clarity, only data with rate ratios between $10^{-3}$ and $10^3$ are shown (>95% of the reactions). (**D**) Rate ratio ($k_{cold}/k_{warm}$) data

*Figure 2 continued on next page*

*Figure 2 continued*

(solid line, $n$ = 2223 from panel C) compared to fold change control distribution (same $T_{Growth}$; dashed line, median = 1.0-fold, 95% CI [0.89, 1.13], $n$ = 615 reactions; p = 0.21, Mann–Whitney U test, two-sided). The black vertical line represents no rate enhancement change with temperature (i.e., rate ratio = 1). (**E, F**) The significance and magnitude of the linear fit of reaction rate as a function of $T_{Growth}$ for negative slopes (E, $n$ = 487) and positive slopes (F, $n$ = 464) in log space. Enzyme Commission (E.C.) number and (substrate) indicated for reactions significantly associated with temperature (Bonferroni correction; p-value < $5.3 \times 10^{-5}$, $n$ = 951). Dotted horizontal lines at p = $-\log_{10}(5.3 \times 10^{-5})$. 5.3.1.1: triose-phosphate isomerase; G3P: glyceraldehyde 3-phosphate; 3.1.1.74: cutinase; 4-NPB: 4-nitrophenyl butyrate.

The online version of this article includes the following source data, source code, and figure supplement(s) for figure 2:

**Source code 1.** Retrieval of updated BRENDA enzyme entries.

**Source code 2.** Analysis of BRENDA enzyme rate constant entries.

**Source code 3.** Control analysis of temperature-matched BRENDA enzyme rate constant entries.

**Source data 1.** Downloaded BRENDA enzyme entries (July 2021).

**Source data 2.** Analyzed BRENDA enzyme entries.

**Figure supplement 1.** Specifying assay temperature and organism optimal growth temperature range per reaction does not alter conclusions.

**Figure supplement 2.** Enzyme rate constant data for $k_{cat}/K_M$ do not indicate rate compensation, supporting the conclusions from the $k_{cat}$ analysis in the main text.

**Figure supplement 3.** Example $m_{rate}$ plots (9 of 951 reactions shown).

---

serine at this position indicates that there are additional factors that influence its rate enhancement (*Figure 3C* and see Discussion).

For KSI, the value of $k_{cat}$ decreased as a function of $T_{Growth}$, but the shallow slope ($m_{rate}$ = –0.006, p = 0.02) (*Figure 3D*) and the small coefficient of determination ($R^2$ = 0.01) of this relationship indicate that $T_{Growth}$ accounts for little of the observed 80-fold rate variation. Similar activity trends were observed at an assay temperature of 15°C (*Figure 3—figure supplement 3*).

## Testing stability compensation using literature data

The absence of evidence for rate compensation led us to reinvestigate the widely accepted relationship between stability and growth temperature. Prior work has shown that the mean temperature optimum of enzymes correlates well with organism $T_{Growth}$ ($r$ = 0.75, *Engqvist, 2018*), but enzyme temperature optima reflect a combination of rate and stability effects. To isolate stability, we surveyed the relationship between stability and $T_{Growth}$ using the ProThermDB, a collection of experimental data of protein and mutant stability (*Nikam et al., 2021*). Across 433 wild-type variants present in this database, we observed a significant relationship between $T_m$ and $T_{Growth}$ (*Figure 4A*, $R^2$ = 0.43, p = $2 \times 10^{-54}$). For the 43 protein families with more than one reported variant, 39 had a higher melting temperature than their cold-adapted counterpart (*Figure 4B*).

## Discussion

Enzymes have been widely posited to adapt to reduced temperature by increasing rate enhancement (*Figure 1A and B*; *Collins and Gerday, 2017*; *D'Amico et al., 2003*; *Siddiqui and Cavicchioli, 2006*; *Zecchinon et al., 2001*). Our results do not support this intuitive and common model as we found that cold-adapted enzyme variants are not generally faster than their warm-adapted counterparts. Even though there was prior evidence for temperature adaptation of the enzyme KSI that is accompanied by rate effects, we found that little of its overall rate variation was accounted for by organismal $T_{Growth}$, suggesting instead that stability is the dominant driving force underlying the previously identified changes. Our observations suggest that enzyme rate enhancement is unlikely to be the primary trait selected for during adaptation to colder environmental temperatures, broadly and in the model system KSI. It has been assumed throughout the literature that cold-adapted enzymes are faster than warm-adapted enzymes, based on the data presented in *Figure 1C and D*; however, our results do not support this model (*Figure 2C and D*).

Perhaps implicit in the expectation that catalysis will increase in cold adaption is the perspective that faster enzymes are better enzymes, with enzymes reacting at the diffusional limit denoted as 'perfect' (*Knowles and Albery, 1977*). However, most enzymes operate well below the diffusional limit (*Bar-Even et al., 2011*), underscoring that an *optimal* reaction rate constant may be different

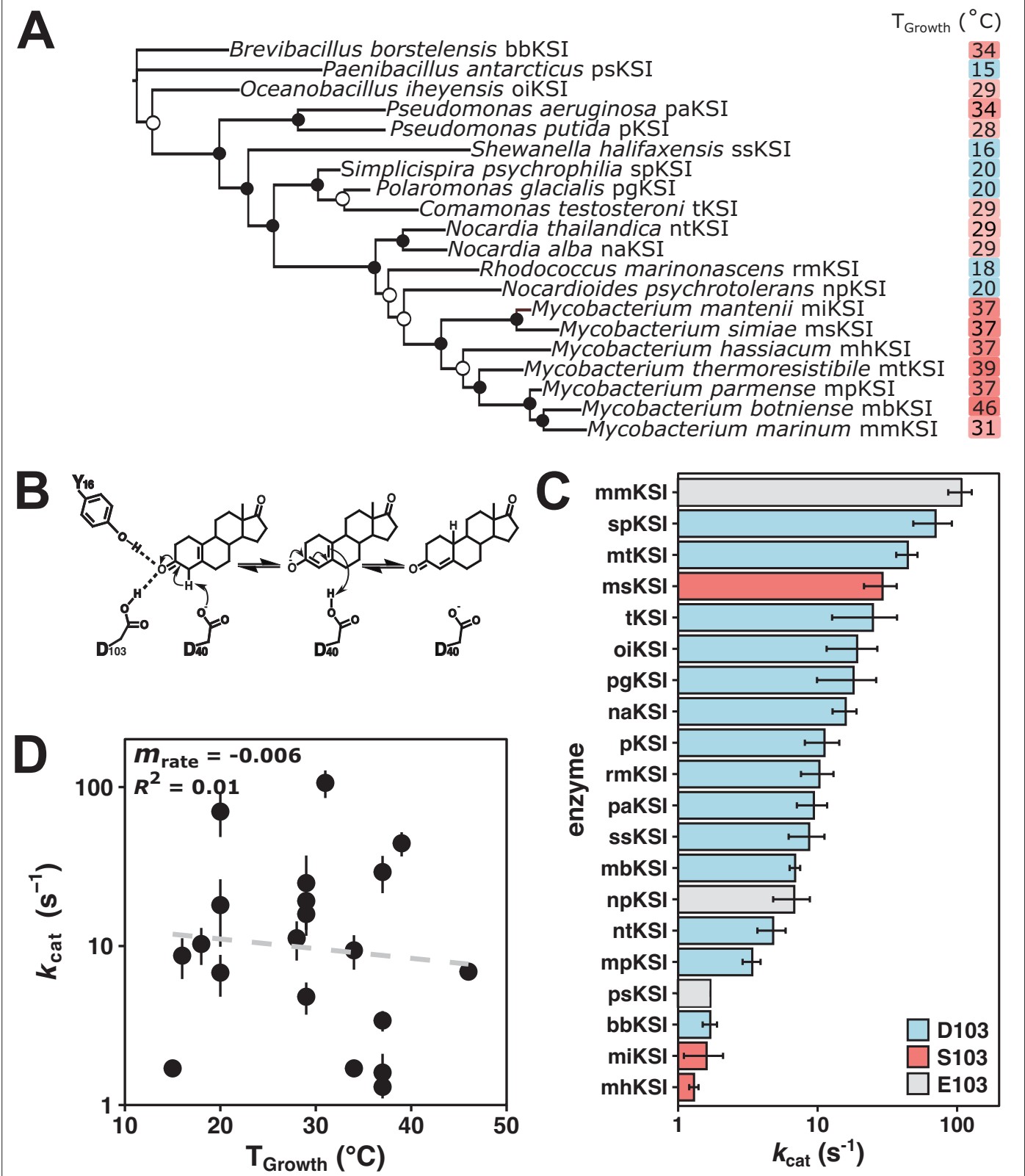

**Figure 3.** Ketosteroid isomerase (KSI) rate enhancements do not indicate rate compensation. (**A**) Unrooted maximum likelihood phylogenetic tree of KSI variants. Closed circles represent bootstrap values of >70%; open circles represent bootstrap values of 40–70%. (**B**) The mechanism of isomerization of the steroid 5 (10)-estrene-3,17-dione by KSI. 5 (10)-EST was used to allow direct measurement of the rate-limiting chemical step $k_{cat}$ (**Pollack et al., 1986**). (**C**) Activity of KSI variants ($k_{cat}$) at a common assay temperature of 25°C. Error bars represent standard deviation of at least two different

*Figure 3 continued on next page*

*Figure 3 continued*

experimental replicates varying [E] at least 5-fold. KSI variants with D103 are represented in blue, S103 in red, and E103 in gray (*Pseudomonas putida* numbering throughout). (**D**) Activity ($\log_{10}(k_{cat})$) of KSI variants at a common assay temperature (25°C) vs. organism growth temperature ($T_{Growth}$) ($n$ = 20, $m_{rate}$ = –0.006, $R^2$ = 0.01, p = 0.02).

The online version of this article includes the following source data and figure supplement(s) for figure 3:

**Source data 1.** Ketosteroid isomerase (KSI) origins and organism growth temperatures.

**Source data 2.** Kinetic measurement of ketosteroid isomerases (KSIs) at 25°C with substrate 5 (10)-estrene-3,17-dione.

**Source data 3.** Kinetic measurement of ketosteroid isomerases (KSIs) at 15°C with substrate 5 (10)-estrene-3,17-dione.

**Figure supplement 1.** ketosteroid isomerase (KSI) variant similarity.

**Figure supplement 2.** ketosteroid isomerase (KSI) variant circular dichroism (CD) spectra are indistinguishable at cold and warm temperature.

**Figure supplement 3.** Ketosteroid isomerase (KSI) rate enhancements vary with organism growth temperature in $k_{cat}$ and in $k_{cat}/K_M$.

than the *maximal* enzyme rate constant. There are multiple reasons why optimal or observed enzyme rate constants may differ from maximal rate constants. Rate optimization in vivo may be accomplished by altering gene expression (*Somero, 2004*), isoform expression (*Somero, 1995*), or cellular pH and osmolytes (*Hochachka and Lewis, 1971*; *Yancey and Somero, 1979*). Alternatively, the optimal enzyme rate may be lower than the maximal rate to channel metabolites and coordinate metabolism (*Prentice et al., 2020*; *Wortel et al., 2018*). Further, models of enzyme-metabolite pathway evolution predict that the subset of enzymes that govern pathway flux through rate-limiting steps are under strong rate selection (*Noda-Garcia et al., 2018*), and it is also possible that maximal enzyme rates are not evolutionarily accessible (*Obolski et al., 2018*). We speculate that rate compensation may be more probable for highly related species that live in similar environments, such as marine species that live at different latitudes or depths but otherwise experience little environmental difference (*Dong and Somero, 2009*).

In contrast to our findings with rate, we observed strong evidence for stability compensation. The temperature dependence of protein unfolding (*Becktel and Schellman, 1987*) may exert a larger

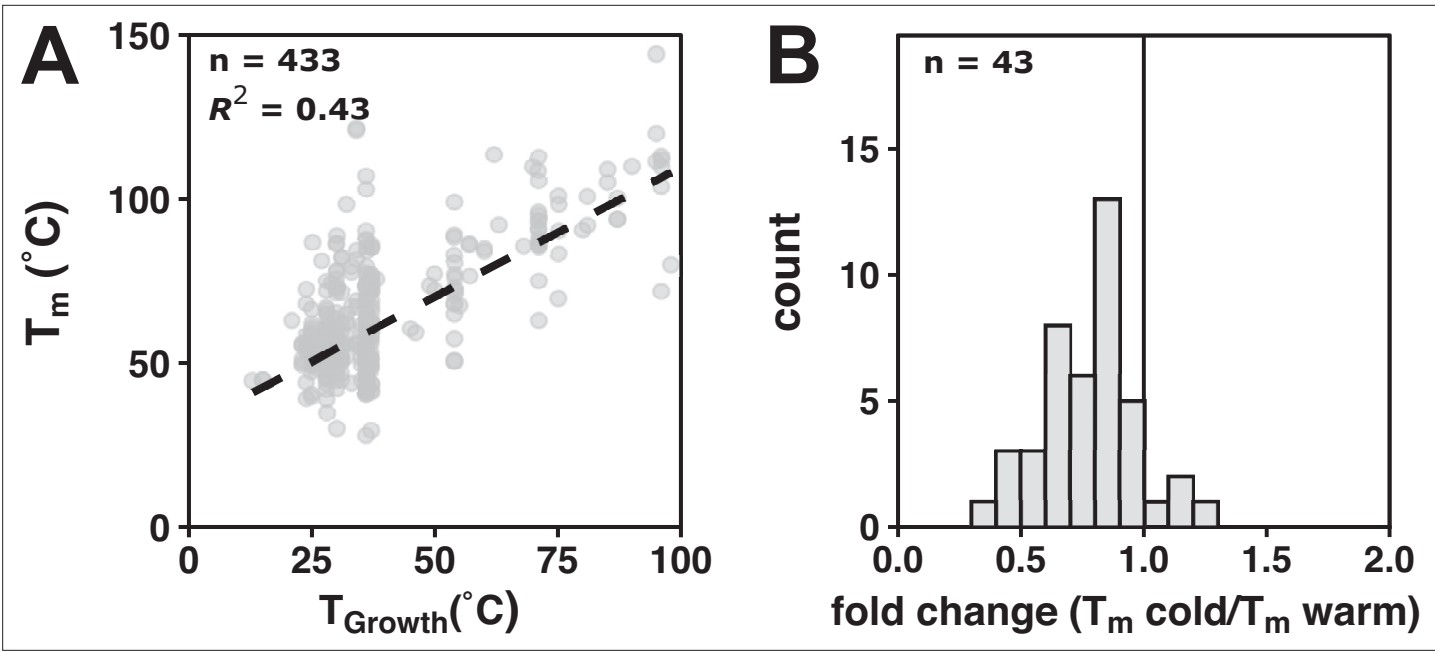

**Figure 4.** Protein stability data display stability compensation. (**A**) Wild-type $T_m$ stability data from ProThermDB as a function of organism $T_{Growth}$. Dashed black line represents a linear fit ($n$ = 433, $R^2$ = 0.43). (**B**) Fold change ($T_m$ cold/$T_m$ warm) of wild-type protein variants ($n$ = 43, median = 0.81, 95% [0.70, 0.85]). The black vertical line represents no change (i.e., fold change = 1).

The online version of this article includes the following source data for figure 4:

**Source data 1.** ProThermDB wild-type protein stability entries.

**Source data 2.** ProThermDB wild-type protein stability entries of protein families with more than one reported variant.

driving force on adaptation than the temperature dependence of rate. There may be an additional strong selection pressure to avoid unfolded states, as misfolded protein has been demonstrated to have deleterious fitness effects (*Geiler-Samerotte et al., 2011*) and cells expend considerable energy to clear misfolded variants using chaperones and degradation pathways (*Clague and Urbé, 2010*; *Hartl et al., 2011*; *Lund, 2001*). Additionally, adaptive paths toward stability may be more abundant and more accessible than analogous paths toward rate enhancement, given that each protein may be stabilized individually through a wide variety of mechanisms (*Hart et al., 2014*) and less constrained by biological context than an enzyme evolving synergistically with complex metabolic networks. The recent discovery of 158,184 positions from 1005 enzyme families that vary with growth temperature may further expand our understanding of the molecular strategies that underlie protein stabilization (*Pinney et al., 2021*).

The observation that KSI variants from high temperature correlate with serine at position 103 may reflect selection for the stabilizing effects of S103 (*Pinney et al., 2021*). In contrast, the observation that one of the fastest KSI variants contains the stabilizing but slowing active site residue (the serine at position 103 in msKSI, *Figure 3C*) may illustrate some of the evolutionary complexity alluded to above. As observed with other KSI variants, the msKSI variant with the wild-type serine residue mutated to aspartic acid (S103D) has increased activity and decreased stability. However, in msKSI, the decreased stability from the S103D mutation renders it partially unfolded even in the absence of denaturants (*Pinney et al., 2021*). This result suggests a model where drift or other factors have led to an overall destabilized scaffold, such that msKSI cannot accommodate the activating S103D change (without unfolding) and has made other as yet unidentified amino acid changes to increase activity.

Flexibility has been posited to mechanistically link rate and stability, with multiple underlying interconnections discussed (see *Supplementary file 1*; *Åqvist et al., 2017*; *Arcus et al., 2016*; *D'Amico et al., 2003*; *Daniel and Danson, 2010*; *Nguyen et al., 2017*; *Saavedra et al., 2018*). Nevertheless, there are many degrees of freedom in an enzyme and most motions are not expected to be coupled to the enzyme reaction coordinate. Our observation of the absence of widespread rate compensation to temperature in contrast to observed stability compensation is consistent with this perspective, as are prior examples of enzyme stabilization in the absence of detrimental rate effects (*Minges et al., 2019*; *Miyazaki et al., 2000*; *Siddiqui, 2017*; *Wintrode and Arnold, 2000*; *Zhao and Feng, 2018*). A more complex relationship between these traits seems likely and underscores the need to relate individual and coupled atomic motions to overall flexibility, catalysis, and stability to unravel their intricate interconnections.

To understand why enzyme properties such as rate and stability measured with purified enzymes vary across organisms, we will need to determine their effects on fitness across biological and environmental contexts, including environments that vary in temperature. Such studies may synergistically deepen our understanding of enzyme function, organismal evolution, and ecosystems.

## Materials and methods

**Key resources table**

| Reagent type (species) or resource | Designation | Source or reference | Identifiers | Additional information |
|---|---|---|---|---|
| Strain, strain background (*Escherichia coli*) | BL21(DE3) | Sigma-Aldrich | CMC0016 | Electrocompetent cells for protein expression |
| Chemical compound, drug | 5 (10)-estrene-3,17-dione | Steraloids | E4500-000 | Ketosteroid isomerase reaction substrate |

### Literature enzyme rate analysis

To capture enzyme rates reported throughout the literature, the BRENDA database was accessed using SOAP July 2021 (*Chang et al., 2021*) (www.brenda-enzymes.org) and the $k_{cat}$ and $k_{cat}/K_M$ database entries retrieved by Enzyme Commission (E.C.) number were parsed for measurement value, substrate, rate, assay temperature, and variant (wild-type or mutant) status (Source Code 1). Microbial optimum growth temperature (*Engqvist, 2018*) values from median organism optimal growth temperatures for microbes in culture ($T_{Growth}$) were matched by organism name to rate entries. (Natural temperatures in the wild may differ. The definition of $T_{Growth}$ used herein is the same as that used in the studies leading to the rate compensation model.) Rate data were filtered for $k_{cat}$ and $k_{cat}/K_M$ values of

wild-type enzymes. Reactions are defined by E.C. number–substrate pair. The median value was taken in the case of multiple measurements of the same enzyme variant with the same substrate.

The rate ratio $k_{cold}/k_{warm}$ per reaction was determined by dividing rate constant of the enzyme from the organism with the minimum $T_{Growth}$ by the rate constant of the enzyme from organism with the maximum $T_{Growth}$. We use the term 'cold-adapted variant' to refer to an enzyme from an organism annotated with lower $T_{Growth}$ values. If a maximum or minimum $T_{Growth}$ was shared between enzyme variants, then the median rate of the two variants was used in the rate ratio calculation. To account for enzyme rate variation arising independently of temperature, a control distribution from reactions with variants of the same $T_{Growth}$ was derived. The fold change of the maximum value over the minimum value $k_{max}/k_{min}$ and its reciprocal $k_{min}/k_{max}$ was calculated for each reaction from the same $T_{Growth}$ with at least two variants. To compare the rate ratio distribution of the data to the rate ratio control, the nonparametric two-sided Mann–Whitney U test was used with a significance threshold of $p < 0.05$. As no temperature-dependent trends emerged when data were restricted to measurements made at 25°C or 37°C or when the $T_{Growth}$ range was limited to $>\Delta 20°C$ and $>\Delta 60°C$, we used all data in the main analysis. We determined confidence intervals of the median parameters of the rate ratio distributions by bootstrap analysis (boot package in R, 10,000 replications) (*Canty and Ripley, 2021*; *Davison and Hinkley, 1997*). The $m_{rate}$ values (slopes) per reaction were calculated by performing a linear regression relating the $\log_{10}$(rate) vs. organism $T_{Growth}$. Significance threshold, corrected for multiple tests, was $p < 5.33 \times 10^{-5}$ (Bonferroni correction; $p < 0.05/951$).

## KSI variant identification, cloning, expression, and purification

Putative KSI variants were identified by sequence relatedness to known KSI variants. Selection of variants was guided by associating putative KSI sequences with $T_{Growth}$ by species (*Engqvist, 2018*). Seventeen variants were synthesized (GenScript or Twist Biosciences) and cloned (Gibson Assembly Protocol, New England Biolabs or Twist Biosciences) into pET-21(+) vectors. KSI variants were aligned using default parameters of Clustal Omega (*Madeira et al., 2019*) and the maximum likelihood tree was constructed using IQ-TREE with default parameters (*Hoang et al., 2018*; *Nguyen et al., 2015*). The constructs were expressed in *Escherichia coli* BL21(DE3) cells and purified as previously described (*Kraut et al., 2010*).

## KSI kinetic measurements

The KSI substrate 5 (10)-estrene-3,17-dione (5 (10)EST) was purchased from Steraloids (Newport, RI). Reactions of purified KSIs with 5 (10)EST were monitored continuously at 248 nm using a Perkin Elmer Lambda 25 UV/Vis spectrometer with an attached VWR digital temperature controlled circulating water bath (*Pinney et al., 2021*). Temperatures within the cuvettes were measured post-reaction using a platinum electrode thermistor (Omega Engineering) and the temperature of the circulating water bath was modified to maintain a constant internal cuvette temperature between reactions. Reactions were conducted in 40 mM potassium phosphate buffer, pH 7.2, 1 mM disodium EDTA, with 2% DMSO as a co-solvent to maintain substrate solubility. The kinetic parameters $k_{cat}$ and $K_M$ were determined by fitting the observed initial velocity of each reaction as a function of 5 (10)EST concentration (9–600 μM; six to seven different substrate concentrations per experiment) to the Michaelis–Menten equation. Reported values of $k_{cat}$ and $K_M$ are the average of three to nine independent experiments with at least two different enzyme concentrations varied by at least 5-fold. Reported errors are the standard deviations of these values.

## KSI CD

CD spectra were collected for each KSI variant in 40 mM potassium phosphate buffer, pH 7.2, 1 mM EDTA, at enzyme concentration 20 μM at 5°C and 25°C. Measurements were made on a J-815 Jasco Spectrophotometer between 190 and 250 nm at 1 nm bandwidth and 50 nm/min scanning speed in a 0.1 cm cuvette (Hellma Analytics).

## Literature stability analysis

Wild-type mutation type stability data were downloaded from ProThermDB (*Nikam et al., 2021*) with the following fields: protein information (entry, source, mutation, E.C. number), experimental conditions (pH, T, measure, method), thermodynamic parameters ($T_m$, state, reversibility), and literature

(PubMed ID, key words, reference, author). Wild-type protein data were matched by species name to microbial optimal growth temperatures $T_{Growth}$ (*Engqvist, 2018*).

## Acknowledgements

We thank M Pinney, H McShea, D Mokhtari, C Markin, IN Zheludev, J Cofsky, EE Duffy, P Harbury, C Khosla, and members of the Herschlag lab for thought-provoking discussions and review of this manuscript. We also thank F Sunden, A Chu, IN Zheludev, and F Yabukarski for experimental assistance and B Eskildsen and D Mokhtari for computational assistance. This research was supported by NSF Grant MCB-1714723, Stanford ChEM-H Chemistry-Biology Interface Training Program, and an NSF Graduate Research Fellowship to CDS and an NSF RET Supplement to JS.

## Additional information

### Funding

| Funder | Grant reference number | Author |
|---|---|---|
| National Science Foundation | Graduate Research Fellowship | Catherine Stark |
| National Science Foundation | MCB-1714723 | Catherine Stark Teanna Bautista-Leung Joanna Siegfried Daniel Herschlag |
| Chemistry, Engineering and Medicine for Human Health, Stanford University | Chemistry-Biology Interface Training Program | Catherine Stark |

The funders had no role in study design, data collection and interpretation, or the decision to submit the work for publication.

### Author contributions

Catherine Stark, Conceptualization, Data curation, Formal analysis, Funding acquisition, Software, Supervision, Visualization, Writing – original draft, Writing – review and editing; Teanna Bautista-Leung, Joanna Siegfried, Investigation; Daniel Herschlag, Conceptualization, Formal analysis, Funding acquisition, Methodology, Project administration, Supervision, Writing – original draft, Writing – review and editing

### Author ORCIDs

Catherine Stark http://orcid.org/0000-0002-0848-3858
Daniel Herschlag http://orcid.org/0000-0002-4685-1973

### Decision letter and Author response

Decision letter https://doi.org/10.7554/eLife.72884.sa1
Author response https://doi.org/10.7554/eLife.72884.sa2

## Additional files

### Supplementary files

- Transparent reporting form
- Supplementary file 1. Overview of proposed molecular models of cold adaptation.
- Supplementary file 2. Ketosteroid isomerase (KSI) sequences.

### Data availability

All data generated or analysed during this study are included in the manuscript and supporting files; Source Data files have been provided for Figures 1, 2, 3, and 4.

The following previously published datasets were used:

| Author(s) | Year | Dataset title | Dataset URL | Database and Identifier |
|---|---|---|---|---|
| Chang A, Jeske L, Ulbrich S, Hofmann J, Koblitz J, Schomburg I, Neumann-Schaal M, Jahn D, Schomburg D | 2021 | BRENDA, the ELIXIR core data resource in 2021: new developments and updates | https://www.brenda-enzymes.org/index.php | The Comprehensive Enzyme Information System, 10.1093/nar/gkaa1025 |

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
