## [Editor Report]

Are enzymes found in organisms that optimally grow at colder temperature are more active than the same enzymes found in organisms that optimally grow at warmer temperatures? Here, an assessment of the catalytic constants for approximately 2200 enzymes (obtained from the BRENDA database) showed no correlation between the relative catalytic activity and the optimum growth temperature. Further support for this conclusion was obtained from the measurement of the catalytic constants from a selection of ketosteroid isomerases from organisms that optimally grow between 15 and 46 degrees centigrade. Therefore, additional factors must dominate the selection pressure used to evolve enzymes of the appropriate catalytic activity. In contrast, growth temperature exerts a significant selective pressure on protein stability.

---

## [Decision Letter]

**Decision letter after peer review:**

Thank you for submitting your article "Systematic investigation of the link between enzyme catalysis and cold adaptation" for consideration by *eLife*. Your article has been reviewed by 3 peer reviewers, one of whom is a member of our Board of Reviewing Editors, and the evaluation has been overseen by Philip Cole as the Senior Editor. The reviewers have opted to remain anonymous.

Essential revisions:

1. Additional discussion regarding the apparent effect of optimal growth temperature and the kinetic constants is warranted for Figures 1C/D in contrast to the data presented in Figures 2C/D.

2. For the data presented in Figure 2C the authors should comment on the apparent fact that the rate constants for catalytic activity for the cold and warm adapted enzyme were not conducted at the same temperature. This issue is further complicated by the issue raised by reviewer 3 with regard to the results for amylase conducted at 3 different temperatures.

3. The revision should also address a supplementary issue raised by reviewer 2 with regard to the definition of optimal growth temperature: What does the optimal growth temperature for an organism tell us about an organisms response to evolutionary pressure in the wild? Organisms generally evolve to grow over a fairly wide range of temperatures, not at a single temperature. Optimal growth temperatures are determined for growth in a defined medium, generally one that favors growth; and, not the more rigorous conditions for growth in the wild.

*Reviewer #1 (Recommendations for the authors):*

1. Figure 1C appears to show that those enzymes from the cold adapted organisms are more catalytically active than those from warm adapted organisms. The authors should be required explain in more detail why the correlations that they observe from the data obtained from BRENDA are more appropriate than the correlations observed by others.

2. The paragraph from lines 188 through 195 is confusing. When the authors write S103D to designate a mutation it is not so clear as to what residue now constitutes the "wild-type" and what is considered the "mutation" when they are discussing msKSI.

3. I was surprised to find that the kcat values for KSI vary by approximately 100. Since the growth temperature apparently not contribute to this difference what other factors would lead to such large differences in these rate constants?

*Reviewer #2 (Recommendations for the authors):*

The manuscript lacks precision in the use of the term rate. For example:

"Prior work has shown that temperature optima for observed enzyme rates.…. Rate has no meaning in this sentence. Temperature optima are determined for plots of temperature against some kinetic parameter, such as kcat.

"The observed rate effects (Figure 1C, Figure 1D)." These Figures describe effects on the rate constant kcat.

"Testing the rate compensation model….." What is meant by "rate compensation model"? Nothing came up in a Google search for this term.

*Reviewer #3 (Recommendations for the authors):*

The title of the paper is not appropriate. The data survey here is not a systematic investigation of the whole question of the relationship of enzyme catalysis to cold adaptation. For example, the focus here is on single enzymes only without consideration of pathway kinetics.

The discussion of existing proposals on temperature dependence of activity is limited and somewhat superficial, particularly with regard to proposals lumped together under the general headings of flexibility ('specific' and 'general'). Previous relevant work of Danson et al., showing that different conformational states can account for observed behavior, should be considered. SI: "Flexibility has been posited to mechanistically link rate and stability, with multiple underlying interconnections discussed" – yes, there have been many such proposals, but most of the references cited here do not do so. Rather, they discuss temperature dependence of rates independent of stability.

Altogether, the presentation is not clear and is confusing. The hypothesis that the authors outline is clear, but the approach is too simplistic. If we take the classic amylase example from Feller and Gerday (cited by the authors), then the ratio of rates at a common temperature for the psychrophile and mesophile would give ratios of >1 for T<37, ~1 at T=37 and <1 for T>37. So it really depends on the common temperature that chosen for the comparison and cannot provide a general explanation.

The data presented are useful, but the accompanying discussion is significantly incomplete. As a whole, this work does not provide significant new insight.

---

## [Author Response]

Essential revisions:1. Additional discussion regarding the apparent effect of optimal growth temperature and the kinetic constants is warranted for Figures 1C/D in contrast to the data presented in Figures 2C/D.

As this comparison –the main finding of this paper– was not clear to the reviewers, we now use language that underscores this comparison in the Discussion:

“It has been assumed throughout the literature that cold-adapted enzymes are faster than warm-adapted enzymes, based on the data presented in Figure 1C and 1D; however, our results do not support this model (Figure 2C, Figure 2D).”

We theorize that multiple studies seeming to support the prior model (which the more extensive data herein do not support) resulted from confirmation bias; we surmise that this was not intentional, as well-meaning and thorough scientists may prioritize communicating positive results that appear to contribute to the understanding of natural phenomena. However, the penetrance of the idea of rate compensation into multiple fields (e.g., enzyme mechanism and enzyme engineering) underscores the necessity of being explicit about our findings. We have worked hard in our presentation to respect the contributions of these different groups while presenting our results clearly, now stating in the Results:

“In summary, the data provide no indication of increase rate enhancements as a consequence of decreasing T_Growth_. […] The prior conclusion of widespread temperature adaptation may have arisen from the use of a small set of enzymes (n = 28; Figure 1C, Figure 1D) or from inadvertent confirmation bias (Nickerson, 1998).”

2. For the data presented in Figure 2C the authors should comment on the apparent fact that the rate constants for catalytic activity for the cold and warm adapted enzyme were not conducted at the same temperature. This issue is further complicated by the issue raised by reviewer 3 with regard to the results for amylase conducted at 3 different temperatures.

Analysis at a common temperature supports the conclusions of this work. We now make it clear that this comparison has been made. More specifically, we begin by analyzing all data and then ask if the trends observed with all data are observed with the subset of data at identical temperatures. We have revised the text in the Results to make this clearer –i.e., that this important analysis (presented in Figure 2*—*figure supplement 1) was carried out and supports our conclusions:

“To assess whether rate constant trends were obscured by mixed assay temperatures or narrow T_Growth_ ranges, we analyzed the distributions of rate ratios separated by assay temperature (25°C or 37°C; Figure 2—figure supplement 1A, 1B) and the rate ratios for data representing wider T_Growth_ ranges (>∆20˚C or >∆60˚C; Figure 2—figure supplement 1C, 1D). No temperature-dependent trends emerged, supporting the above conclusion of an absence of widespread rate compensation.”

For the amylase measurements, this is a case where the enzyme rate dependence on temperature differs between three amylase variants (D’Amico et al., 2003). Rate enhancement measurements spanning the larger family of amylases will test whether the observed pattern generally holds across amylase variants. There may be individual instances of more complex behaviors or more patterns to be uncovered in particular enzyme types or families, and we look forward to future research addressing these questions. We did not observe significantly different trends for the 20 KSI variants we investigated at 15 vs. 25°C. In this manuscript, to be clear, our overall ais to address the question and test the model that rate adaptation to temperature is a phenomenon that is general to enzymes. By looking systematically at the available data, we found that the data did not support this model that was previously proposed based on a limited number of published examples.

3. The revision should also address a supplementary issue raised by reviewer 2 with regard to the definition of optimal growth temperature: What does the optimal growth temperature for an organism tell us about an organisms response to evolutionary pressure in the wild? Organisms generally evolve to grow over a fairly wide range of temperatures, not at a single temperature. Optimal growth temperatures are determined for growth in a defined medium, generally one that favors growth; and, not the more rigorous conditions for growth in the wild.

This is an interesting (and complex) question but is not directly relevant to our analyses and results. In this manuscript, we are using the definition of growth temperature that was used previously in the literature and used to derive the general model that we present and test; given that our goal was to test this prior model –and in particular, whether it was supported by data when all available data were included– it was important to maintain the prior definition of growth temperature (e.g., Engqvist, 2018).

We have clarified this in the Materials and methods:

“Natural temperatures in the wild may differ. The definition of T_Growth_ used herein is the same as that used in the studies leading to the rate compensation model.”

Further, if the proposed activity-growth temperature trend were general, we would still expect to see it if the temperature values represented an average.

Nevertheless, this reviewer brings up fascinating evolutionary considerations. Models for temperature adaptation through rate or other adaptation should consider and address other environmental differences including the range of temperatures that an organism experiences, as we have noted in the Discussion:

“To understand why enzyme properties such as rate and stability measured with purified enzymes vary across organisms, we will need to determine their effects on fitness across biological and environmental contexts, including environments that vary in temperature.”

Reviewer #1 (Recommendations for the authors):1. Figure 1C appears to show that those enzymes from the cold adapted organisms are more catalytically active than those from warm adapted organisms. The authors should be required explain in more detail why the correlations that they observe from the data obtained from BRENDA are more appropriate than the correlations observed by others.

We have clarified this in Essential Revision #1 above. Briefly, the prior data were based on a limited number of anecdotal studies (n = 28). In contrast, the BRENDA data we curated represents over 2200 enzymatic reactions, a collation of approximately 100-fold more data than the prior combined studies. Our analysis includes all of the data that are published and catalogued, without regard to whether the system under examination is behaving in the predicted manner. We have emphasized that approach is less biased and is expected to be more representative.

2. The paragraph from lines 188 through 195 is confusing. When the authors write S103D to designate a mutation it is not so clear as to what residue now constitutes the "wild-type" and what is considered the "mutation" when they are discussing msKSI.

We have indicated that serine at position 103 is the wild-type residue in the Discussion:

“The observation that KSI variants from high temperature correlate with serine at position 103 may reflect selection for the stabilizing effects of S103 (Pinney et al., 2021). […] As observed with other KSI variants, the msKSI variant with the wild-type serine residue mutated to aspartic acid (S103D) has increased activity and decreased stability.”

3. I was surprised to find that the kcat values for KSI vary by approximately 100. Since the growth temperature apparently not contribute to this difference what other factors would lead to such large differences in these rate constants?

We were surprised as well. After obtaining this result, we realized that there was little data about how enzyme families vary in reaction parameters. We observe that the median fold change (k_max_/k_min_) is about 10-fold (black line), as compared to the fold change of approximately 100-fold of KSI (blue line) with a wide range of distribution (Author response image 1) .

**Author response image 1. sa2fig1:** Differences in rate constants between enzyme variants. (A) The fold change in *k*_cat_ (k_max_/k_min_) of enzyme reactions (n = 2223). The black line indicates the fold change median value (median *k*_max_/*k*_min_ = 11.8) and the blue line indicates the fold change value of KSI (*k*_max_/*k*_min_ = 82). (B) The fold change in *k*_cat_ (k_max_/k_min_) of enzyme reactions measured at 37°C (n = 319). The black line indicates the fold change median value (median *k*_max_/*k*_min_ = 3.7) and the blue line indicates the fold change value of KSI (*k*_max_/*k*_min_ = 82); while the median is much smaller there remain a number of enzymes with fold change values similar or larger than the KSIs.

We agree that identifying other factors that affect rates is an important area of future study. For example, how much do differential selective pressure of metabolic networks lead to different optimized rates? Addressing the question of how differential selective pressures affect rate constants will require systematic studies of enzyme variances across multiple organisms, as we note in the Discussion (and above):

“To understand why enzyme properties such as rate and stability measured with purified enzymes vary across organisms, we will need to determine their effects on fitness across biological and environmental contexts, including environments that vary in temperature.”

Reviewer #2 (Recommendations for the authors):The manuscript lacks precision in the use of the term rate. For example:"Prior work has shown that temperature optima for observed enzyme rates.…. Rate has no meaning in this sentence. Temperature optima are determined for plots of temperature against some kinetic parameter, such as kcat."The observed rate effects (Figure 1C, Figure 1D)." These Figures describe effects on the rate constant kcat."Testing the rate compensation model….." What is meant by "rate compensation model"? Nothing came up in a Google search for this term.

Thank you for these specific comments. We have clarified the terminology around rate throughout the manuscript. For rate compensation, we have clarified that this is a term that we have introduced in the Introduction.

Reviewer #3 (Recommendations for the authors):The title of the paper is not appropriate. The data survey here is not a systematic investigation of the whole question of the relationship of enzyme catalysis to cold adaptation. For example, the focus here is on single enzymes only without consideration of pathway kinetics.

Systematic is defined as “methodical in procedure or plan” or “marked by thoroughness and regularity” (Merriam-Webster). Our title references that our data survey in this work is systematic with respect to existing data.

The discussion of existing proposals on temperature dependence of activity is limited and somewhat superficial, particularly with regard to proposals lumped together under the general headings of flexibility ('specific' and 'general'). Previous relevant work of Danson et al., showing that different conformational states can account for observed behavior, should be considered. SI: "Flexibility has been posited to mechanistically link rate and stability, with multiple underlying interconnections discussed" – yes, there have been many such proposals, but most of the references cited here do not do so. Rather, they discuss temperature dependence of rates independent of stability.

Danson et al. address how enzyme rates ‘drop off’ with temperature at temperatures lower than needed to unfold them. Their model –of multiple states with differential activity– provides a very reasonable explanation. This behavior though is distinct from the topic of our work –which is a comparison of rate constants for different enzymes from organisms with different growth temperatures, not the same enzyme *vs.* temperature. Regardless, we appreciate that Reviewer #3 has brought this work to our attention and we have included it in the revised manuscript.

Altogether, the presentation is not clear and is confusing. The hypothesis that the authors outline is clear, but the approach is too simplistic. If we take the classic amylase example from Feller and Gerday (cited by the authors), then the ratio of rates at a common temperature for the psychrophile and mesophile would give ratios of >1 for T<37, ~1 at T=37 and <1 for T>37. So it really depends on the common temperature that chosen for the comparison and cannot provide a general explanation.

As noted above, the reported data on amylase is a case (with n = 3 variants) where we know the enzyme activity depends on temperature in different ways for the different variants. The benefit of using broad data is that we are able to test the generality of a hypothesis across different enzyme families and more variants (please see Essential Revision #1 above). We also emphasize that we do see the same absence of a trend when carrying out the analysis at 25°C or 37°C (Figure 2—figure supplement 1).

The data presented are useful, but the accompanying discussion is significantly incomplete. As a whole, this work does not provide significant new insight.

The conclusion of our paper is that a general trend predicted by a widely accepted model is not supported by the data. This is significant, as it disproves a widely accepted trend and especially as there has been much discussion and models already in the literature based on the assumption that this trend is correct. We felt it best to mainly present this new result, point out complexities in selection that might lead to the absence of a general trend, but not speculate about topics not directly linked to the current results.